# MEDTVT-R1: A MULTIMODAL LLM EMPOWERING MEDICAL REASONING AND DIAGNOSIS

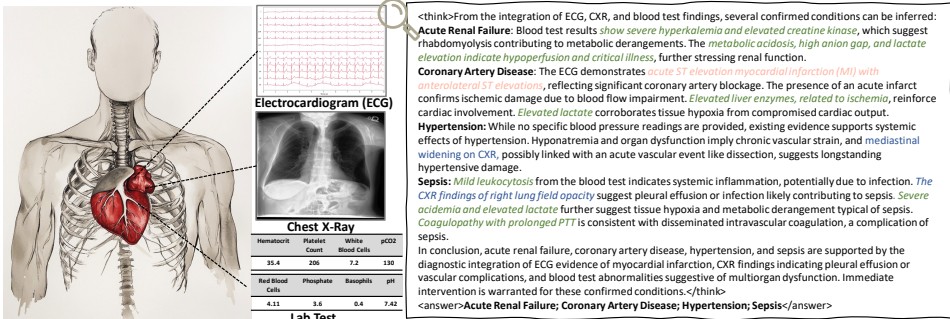

Figure 1: Overview of **MedTVT-R1**: MedTVT-R1 seamlessly integrates Electrocardiogram (**T**ime Series), Chest X-ray (**V**isual Image), and Blood Test (**T**abular Data) to deliver comprehensive long-text **Med**ical reasoning and diagnosis across various diseases.

## ABSTRACT

Accurate and interpretable multi-disease diagnosis remains a critical challenge in medical research, particularly when leveraging heterogeneous multimodal medical data. Current approaches often rely on single-modal data, limiting their ability to comprehensively understand complex diseases. To address this, we propose MedTVT-R1, a novel Multimodal Large Language Model (MLLM) framework designed to integrate clinical multimodal data for reasoning and diagnosing multiple diseases. We construct MedTVT-QA, a curated instruction dataset that provides question-answer pairs for physiological-level interpretations and disease-level diagnoses with a Chain of Evidence approach. MedTVT-R1 incorporates a modality perception layer to capture inter-modal dependencies and adaptively weight modality contributions. Additionally, we employ Group Relative Policy Optimization (GRPO)-based Reinforcement Fine-Tuning with a Jaccard Reward function to enhance diagnostic reasoning. Experimental results demonstrate MedTVT-R1's superiority in multimodal feature utilization and multi-disease diagnosis, offering significant potential for clinical applications such as diagnostic report generation and comorbidity reasoning. The dataset and code will be available on GitHub.

## 1 INTRODUCTION

The rapid development of artificial intelligence (AI) has profoundly reshaped the landscape of medical research and clinical practice, especially in demonstrating significant progress and potential in medical data analysis (Çallı et al., 2021; Liu et al., 2021; Hernandez et al., 2022; Sumon et al., 2025) and disease diagnosis (Cassar et al., 2009; Ghaffar Nia et al., 2023), with an extensive impact (Elstein, 2004).

At present, most existing studies primarily rely on single-modal medical data to perform disease diagnosis (Chen et al., 2024b; Hernandez et al., 2022; Yao et al., 2024; Ansari et al., 2023). Although these single-modal approaches demonstrate certain effectiveness within their respective specific domains, their perception of physiology is often too limited to offer a holistic and comprehensive understanding of complex diseases. Taking diabetes as an example, its physiological manifestations are typically reflected across multiple modalities, such as altered heart rate variability in electrocardiograms (ECG), pulmonary complications observable in chest X-rays (CXR), and abnormal glucose or lipid levels revealed by laboratory blood tests (LAB) (Lin et al., 2021). Therefore, to address

the risk of incomplete or inaccurate diagnoses resulting from reliance on a single modality, it is essential to integrate multimodal medical data for comprehensive and in-depth analysis of complex diseases (Alcaraz & Strodthoff, 2024; Steyaert et al., 2023).

Consequently, there are a number of efforts that have emerged to explore leveraging multimodal medical data for disease diagnosis (Kline et al., 2022; Steyaert et al., 2023; Venugopalan et al., 2021; Abdelaziz et al., 2021). Nevertheless, these methods often make only simple and direct determinations about the presence or absence of a specific disease (Gundapaneni et al., 2024; Kumar, 2022), but struggle with performing robust long-text diagnostic reasoning and generating interpretable clinical insights for multiple diseases, which severely hinders their practical application.

Recently, multimodal large language models (MLLMs) (Zhang et al., 2023; Li et al., 2023a; Liu et al., 2023; 2024c; Tian et al., 2025; Wu et al., 2024) have undergone rapid development and achieved impressive results in a variety of tasks, such as vision-language and audio-language tasks. They have demonstrated strong capabilities in integrating, generalizing, and reasoning across diverse data modalities, offering promising potential for generating interpretable disease diagnosis reports from medical data. Although several pioneering studies have made preliminary attempts to apply MLLMs in the medical field, such as for ECG analysis (Zhao et al., 2024; Tian et al., 2024) or medical image reporting (Shentu & Al Moubayed, 2024; Liu et al., 2024a; Tanno et al., 2025) tasks, these works are still limited to single modalities (*e.g.*, ECG, CXR) and remain at physiological-level understanding rather than disease-level reasoning. Therefore, an MLLM that can perceive and integrate heterogeneous multimodal medical data, thereby enabling interpretable multi-disease reasoning and diagnosis, remains a significant gap in current research.

From above observations, we propose **MedTVT-R1**—an MLLM leveraging clinical multimodal data's complementarity/corroboration for multi-disease reasoning/diagnosis (advances in Figure 1). To this end, we construct **MedTVT-QA**—the first instruction dataset covering three modalities (ECG, CXR, LAB) with QA pairs. It includes physiological interpretations and disease-level diagnoses via a Chain of Evidence (CoE) (using cross-modal complementarity/corroboration), laying a foundation for MLLMs' progressive multimodal integration for physiological perception/multi-disease diagnosis. We also add a Modality Perception Layer (MPL) to MedTVT-R1, capturing cross-modal dependencies and adaptively weighting modality contributions by disease relevance to maximize cross-modal interaction/information use. Inspired by DeepSeek-R1 (Guo et al., 2025), we adopt Reinforcement Fine-Tuning (RFT) via Group Relative Policy Optimization (GRPO) for post-training, with a Jaccard Reward function for multi-disease scenarios—boosting reasoning capability. Extensive experiments show MedTVT-R1's superiority in single-modality physiological understanding and multimodal disease diagnosis, with implications for clinical MLLM applications (e.g., interpretable diagnostic reports, complex comorbidity reasoning). Our contributions are summarized as follows:

- We introduce MedTVT-QA, the first medical instruction dataset that features heterogeneous modalities including ECG (Time Series), CXR (Visual Images), and LAB (Tabular Data).
- We propose MedTVT-R1, a novel MLLM framework that fully leverages the complementarity and mutual corroboration among clinical multimodal data for interpretable diagnosis of complex comorbidities.
- We employ a Reinforcement Fine-Tuning (RFT) strategy based on Group Relative Policy Optimization (GRPO) incorporating a dedicated Jaccard reward function to unlock data potential and enhance the model's reasoning accuracy.
- Extensive experiments demonstrate that MedTVT-R1 achieves state-of-the-art performance in physiological representation understanding across various modalities and multimodal diagnosis and report generation for comorbidity.

## 2 RELATED WORK

**MLLM for Medical Diagnosis.** The application of Multimodal Large Language Models (MLLMs) in medical diagnosis has gained significant attention due to their ability to process and integrate diverse data modalities, such as text (Li et al., 2025; Liévin et al., 2024; Jin et al., 2024; Gallifant et al., 2025), images (Irvin et al., 2019; Lee et al., 2025; 2023; Lu et al., 2024b), and tabular data (Bisercic et al., 2023; Huang et al., 2024). Early works focused on single-modal approaches, such as text-based models for clinical note analysis (Jin et al., 2024; Yuan et al., 2024; Liévin et al., 2024), image-based models for radiology interpretation (Lee et al., 2025; Irvin et al., 2019), or ECG-based models

for cardiac status analysis (Zhao et al., 2024; Yu et al., 2023; Lan et al., 2025; Yang et al., 2025). Although significant advancements have been made, existing research has yet to integrate **T**ime series data (e.g., Electrocardiograms), **V**isual data (e.g., chest X-rays), and **T**abular data (e.g., lab results) into a unified framework for comprehensive **Med**ical disease analysis and diagnosis. To bridge this gap, we introduce **MedTVT-R1**, a multimodal large language model designed to seamlessly combine CXR, ECG, and lab data through cross-modal interactions and contribution-aware operator, enabling accurate and interpretable disease diagnosis.

**Reinforcement Learning with Verifiable Rewards.** Group Relative Policy Optimization (GRPO) (Guo et al., 2025), unlike Proximal Policy Optimization (PPO) (Yu et al., 2022; Schulman et al., 2017) which estimates advantages through a reward model, approximates advantages by obtaining multiple samples from the LLM using the same prompt, with the advantage being the normalized reward for each response within its set of generated responses, achieving notable success in text-based tasks (Shao et al., 2024; Ramesh et al., 2024; Dao & Vu, 2025) such as summarization and dialogue generation, as well as vision tasks (Liu et al., 2025; Tan et al., 2025) like image captioning. Recently, GRPO has been applied to medical image analysis (Lai et al., 2025; Pan et al., 2025). However, it has not yet been utilized for multimodal tasks in the crucial area of multi-disease diagnosis, which requires the integration of text, images, time series, and tabular data. In this work, we are pioneering the application of GRPO with a newly designed reward function, the Jaccard Reward, to enhance the accuracy of multi-disease prediction.

# 3 METHODOLOGY

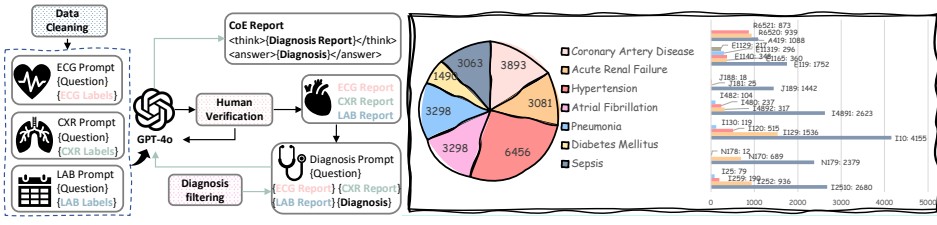

(a) Pipeline of Dataset Construction   (b) Disease Distribution of Training Data

Figure 2: MedTVT-QA dataset construction and disease distribution. (a) Pipeline of Dataset Construction: labels are refined to ensure consistency, prompts guide GPT-4o in generating verified physiological-level reports, which are combined with diagnostic labels to produce disease-level reports. Diagnostic labels are organized into seven primary categories with detailed subtypes. (b) Disease distribution of MedTVT-QA, with subtypes classified by ICD-10 codes. More details can be found in Appendix B.

## 3.1 MEDTVT-QA

To enable MLLMs to perform physiological understanding and disease diagnosis from heterogeneous multimodal medical data, we curate patient-level ECG readings, blood test results, and chest X-ray (CXR) images from the MIMIC-IV family of datasets (Johnson et al., 2020; Gow et al., 2023; Johnson et al., 2019), facilitated by Symile (Saporta et al., 2024). All modalities are acquired from the same individuals within a clinically proximate period during hospitalization, yielding 8,706 aligned multimodal combinations (8,331 for training, 375 for testing). Additional diagnostic labels are sourced from MIMIC-IV-ECG-EXT-ICD (Strodthoff et al., 2024). These data underpin the construction of multimodal question–answer (QA) pairs for both physiological representation analysis and disease-level diagnostic reasoning, as illustrated in Figure 2(a).

**QA Pairs on Physiological-level Representation Analysis.** To enable MLLMs to acquire a basic comprehension of the physiological meanings represented by each modality, *i.e.*, ECG, CXR, and LAB, and to lay the groundwork for subsequent multi-disease reasoning and diagnosis based on multimodal data, we first construct QA pairs focused on physiological-level analysis for each modality. Specifically, taking the ECG modality as an example, we design a rigorous prompt with reference to physiological-level annotations, which consists of Role Setting, Task Description, Answer Guidance, and Answer Format to instruct GPT-4o (Achiam et al., 2023), as outlined below:

> *Role Setting*: You are a renowned cardiologist with expertise in interpreting electrocardiograms (ECGs).
> *Task Description*: The ECG analysis has yielded the following labels: {labels}. Based on these labels, please address the question: {question}.
> *Answer Guidance*: Your response should incorporate all relevant labels, excluding any unrelated ones. Provide a synthesis of the labels, focusing on clinical significance.
> *Answer Format*: Begin with a brief introduction to your analysis. Provide detailed explanations for each specific ECG label. Offer a concise summary.

Here, {labels} refers to physiological-level annotations for the ECG data, such as Sinus Rhythm or Left Bundle Branch Block. {question} is randomly sampled from several carefully crafted question templates, such as 'What information can be derived from the ECG?'. An example of a constructed QA pair that summarizes and analyzes the physiological representations encompassed in ECG is:

> **Question**: What information can be derived from the ECG ¡ecg¿?
> **Answer**: ECG Analysis: Upon evaluating the electrocardiogram (ECG), the following observations can be made:
> 1. Sinus Rhythm: The heart is under the control of the sinoatrial (SA) node, which is the natural pacemaker. This finding suggests that the electrical activity of the heart is originating from the SA node and progressing through the normal conduction pathway, leading to a regular rhythm. This is a foundational indicator of organized cardiac activity.
> 2. Left Bundle Branch Block (LBBB): There is evidence of delayed or disrupted conduction through the left bundle of the heart's electrical conduction system. This results in a characteristic pattern on the ECG, with a widened QRS complex (¿120 ms) and specific morphologies in leads, such as a broad, notched appearance in V5-V6 and an absence of normal septal depolarization. LBBB may occur in conditions like ischemic heart disease, cardiomyopathy, or hypertension-induced structural changes, though it can also occasionally be found in asymptomatic individuals.
> Summary: The heart demonstrates a sinus rhythm, indicating normal initiation of electrical activity. However, the presence of a left bundle branch block signifies an abnormality in electrical conduction, often associated with underlying cardiac pathology. As a whole, this ECG is classified as abnormal due to the conduction disturbance, warranting further investigation to assess structural or functional cardiac issues.

Similar prompts are also applied to CXR and LAB data to organize the corresponding physiological-level QA pairs for each modality. It is worth noting that, for LAB data, we group 50 common laboratory indicators into seven categories according to their physiological significance to facilitate processing. The content generated above for physiological-level representation analysis undergoes manual review and revision by professionals to ensure its rationality and reliability. More details on physiological-level annotation and examples of QA pairs for each modality can be found in the Appendix A to C.

**QA Pairs on Disease-level Diagnostic Reasoning.** Building upon the aforementioned completed physiological-level representation analysis for each modality, we further construct QA pairs that fully integrate information across modalities and conduct disease-level diagnostic reasoning, thereby enhancing the capability of MLLMs to handle complex multiple diseases. We focus on seven common and clinically significant diseases for which supporting evidence can be found in ECG, CXR, and LAB data, including *Coronary Artery Disease, Acute Renal Failure, Hypertension, Atrial Fibrillation, Pneumonia, Diabetes Mellitus,* and *Sepsis*, each of which contains several subtypes, with details in the Appendix B. The corresponding statistics are presented in Figure 2 (b). We also employ a four-element prompt with reference to disease-level annotations to instruct GPT-4o, and compel its response to include a Chain of Evidence (CoE) to fully leverage the complementarity and mutual corroboration among modalities, thereby thoroughly extracting multimodal evidence for disease diagnosis, as follows:

> **Role Setting**: You are a renowned diagnostician with expertise in integrating ECG, CXR, and blood test results.
> **Task Description**: The following diagnostics have been provided:
> • ECG Analysis: {ecg_report}
> • CXR Analysis: {cxr_report}
> • Blood Test Analysis: {blood_test_report}
> • Diseases: {result_diseases}
> You need to pretend that the ECG, CXR, and blood test analyses are based on your interpretation of the raw data, and the final diagnosis is your synthesis of these three diagnostic methods, please address the question: {question}
> **Answer Guidance**: Please find definitive evidence from the ECG, CXR, and blood test results, leveraging the complementarity and mutual corroboration of these three modalities, to robustly prove the reasons why the patient has the diseases I provided. Your response must include every disease I provided, using the exact wording I provided, and you must not mention any diseases other than those I provided. Please make sure to provide evidence for these diagnoses! These are confirmed conditions.
> **Answer Format**: <think>{Diagnostic evidence synthesized from the three modalities}</think>\n <answer>{disease1; disease2; ...}</answer>

Here, {ecg_report}, {cxr_report}, and {blood_test_report} respectively represent the physiological-level analyses of the three modalities. {result_diseases} refers to the disease-level annotation of the sample. {question} is randomly sampled from several carefully crafted question templates, such as 'Can you analyze my ECG, CXR and lab result to determine my probable conditions?'. The CoE is implemented by 'Please find definitive evidence...'. The content obtained in this process is also reviewed by professionals to enhance its trustworthiness. An example of a QA pair that integrates multimodal information to mine evidence for multi-disease reasoning and diagnosis is shown in Figure 3, and the complete version is provided in the Appendix E.

## 3.2 MEDTVT-R1

Based on the meticulously constructed MedTVT-QA dataset described above, we propose MedTVT-R1, an MLLM framework capable of fully exploiting the complementarity and mutual corroboration of multimodal medical data for interpretable multi-disease reasoning and clinical diagnosis. In the following, we will introduce the model architecture and training strategy of MedTVT-R1 in detail.

Figure 3: Pipeline of MedTVT-R1. Pretraining processes ECG, CXR, and LAB data through encoders and projectors, combined with prompts, to train projectors and LLM's LoRA for enhanced physiological understanding. The SFT stage adds a Modality Perception Layer for interaction and integration, refining disease analysis. The RFT stage applies GRPO, using the SFT-trained model for policy and inference, optimizing KL divergence and reward loss.

### 3.2.1 ARCHITECTURE

The proposed MedTVT-R1 mainly consists of modality-specific encoders and projectors, a Modality Perception Layer (MPL), and a Large Language Model (LLM), with its overall architecture illustrated on the left side of Figure 3. Given the raw data of ECG signals $\mathbf{X_E} \in \mathbb{R}^{N \times L}$, CXR images $\mathbf{X_C} \in \mathbb{R}^{C \times H \times W}$, and LAB tables $\mathbf{X_L} \in \mathbb{R}^{N'}$, they are first processed by their respective modality-specific encoders for feature extraction, and then the encoded features are fed into modality-specific projectors to a shared dimension $d$ for alignment and compatibility with the textual embedding space of the LLM, facilitating seamless integration between multimodal features and textual tokens; this process can be formulated as follows:

$$\mathbf{Z_E} = g_{\mathrm{E}}(f_{\mathrm{E}}(\mathbf{X_E})) \in \mathbb{R}^d, \quad \mathbf{Z_C} = g_{\mathrm{C}}(f_{\mathrm{C}}(\mathbf{X_C})) \in \mathbb{R}^d, \quad \mathbf{Z_L} = g_{\mathrm{L}}(f_{\mathrm{L}}(\mathbf{X_L})) \in \mathbb{R}^d, \quad (1)$$

where $\mathbf{Z_{E/C/L}}$ denotes the projected multimodal features, and $f_{\mathrm{E/C/L}}$ and $g_{\mathrm{E/C/L}}$ represent the modality-specific encoders and projectors, respectively.

Subsequently, to enable efficient interaction and fusion among modalities, we introduce a Modality Perception Layer (MPL), which comprises a Cyclic Multi-Head Attention (CMHA) mechanism and a Contribution-Aware Operator (CAO). Specifically, the projected features $\mathbf{Z_E}$, $\mathbf{Z_C}$, and $\mathbf{Z_L}$ are first processed by the CMHA mechanism, in which each modality feature cyclically serves as the Query, Key, and Value to compute multi-head attention, enabling comprehensive capture of cross-modal dependencies and facilitating in-depth information exchange among ECG, CXR, and LAB features. After one round of cycling, the outputs are fused through average pooling, while a residual connection is employed to preserve modality-specific information. This process can be formulated as follows:

$$\mathbf{F} = \mathrm{AveragePooling}(\mathrm{CMHA}(\mathbf{Z_E}, \mathbf{Z_C}, \mathbf{Z_L})), \quad \mathbf{M_{E/C/L}} = \mathbf{Z_{E/C/L}} + \mathbf{F}, \quad (2)$$

where $\mathbf{M_{E/C/L}}$ denotes the updated features of each modality, which encapsulate both modality-specific and modality-shared information. Recognizing that each modality contributes in varying degrees to the reasoning and diagnosis of various diseases, for example, ECG features are relatively more important for detecting Coronary Artery Disease, we design a Contribution-Aware Operator that adaptively assigns weights to the features of each modality based on the diagnostic context, which can be formulated as follows:

$$\mathbf{T_E}, \mathbf{T_C}, \mathbf{T_L} = \sigma(h[\mathbf{M_E} : \mathbf{M_C} : \mathbf{M_L}]) \otimes (\mathbf{M_E}, \mathbf{M_C}, \mathbf{M_L}), \quad (3)$$

where [:] denotes the concatenation operation, $h$ is a learnable transformation matrix, $\sigma$ represents the Sigmoid activation, and $\otimes$ denotes element-wise multiplication. The final multimodal features $\mathbf{T_E}$, $\mathbf{T_C}$, and $\mathbf{T_L}$ are used to replace the placeholders `<ecg>`, `<cxr>`, and `<lab>` in the text tokens, which are obtained by processing the input prompt through the tokenizer and embedding layer. An example input prompt could be: 'What illnesses might be indicated by the findings from my ECG `<ecg>`, CXR `<cxr>`, and blood work `<lab>`?'. The resulting sequence $\mathbf{T}_{\mathrm{input}} = \{\mathbf{T_Q}, \mathbf{T_E}, \mathbf{T_C}, \mathbf{T_L}, \mathbf{T_A}\}$ is then fed into the LLM, where $\mathbf{T_Q}$ and $\mathbf{T_A}$ are derived from the QA pairs in the MedTVT-QA dataset.

### 3.2.2 TRAINING STRATEGY

We employ a three-stage training strategy for MedTVT-R1, which includes Pre-training (PT), Supervised Fine-Tuning (SFT), and Reinforcement Fine-Tuning (RFT), to progressively enhance its ability to perceive the physiological representations of each modality and integrate multimodal information for interpretable multi-disease reasoning and diagnosis.

**Pre-training.** With the aim of helping the model form an initial understanding and awareness of the physiological significance across all modalities, we first perform pre-training using physiological-level QA pairs from the MedTVT-QA dataset. During this stage, the projectors and the Low-Rank Adaptation (LoRA) modules embedded in the LLM are set as trainable, while the other components remain frozen. Notably, the MPL module is absent at this stage as no cross-modal interaction is involved. The optimization objective is to maximize the likelihood of generating the target response tokens, formalized as:

$$\mathcal{L}_{\text{PT}} = -\mathbb{E}_{(\mathbf{T_Q}, \mathbf{T_{E/C/L}}, \mathbf{T_A}) \sim \mathcal{D}} \sum_{t=1}^{T} \log \pi_\theta(y_t \mid \mathbf{T_Q}, \mathbf{T_{E/C/L}}, y_{<t}), \tag{4}$$

where $\pi_\theta(y_t \mid \cdot)$ denotes the conditional probability of generating the $t$-th token $y_t$, given the prompt, modality features, and the previously generated tokens $y_{<t}$.

**Supervised Fine-Tuning.** With the pretrained model that already demonstrates a solid understanding of the physiological significance of each modality, we further conduct SFT based on disease-level QA pairs with CoE logic from the MedTVT-QA dataset to equip the model with the capability to synthesize multimodal representations and uncover the complementarity and mutual corroboration among modalities for multi-disease reasoning and diagnosis. During this stage, the MPL and the LoRA modules embedded in the LLM are set to be trainable while the other components remain frozen, and the optimization objective is similar to that of the pre-training stage, namely:

$$\mathcal{L}_{\text{SFT}} = -\mathbb{E}_{(\mathbf{T_Q}, \mathbf{T_E}, \mathbf{T_C}, \mathbf{T_L}, \mathbf{T_A}) \sim \mathcal{D}} \sum_{t=1}^{T} \log \pi_\theta(y_t \mid \mathbf{T_Q}, \mathbf{T_E}, \mathbf{T_C}, \mathbf{T_L}, y_{<t}). \tag{5}$$

**Reinforcement Fine-Tuning.** To unlock the potential of the constructed dataset and boost the model's reasoning performance, inspired by the advancements of DeepSeek-R1, we perform RFT using Group Relative Policy Optimization (GRPO) under the Reinforcement Learning with Verifiable Rewards (RLVR) framework. The training corpus and trainable components remain consistent with those in the SFT stage. The optimization objective can be formulated as:

$$\max_{\pi_\theta} \mathbb{E}_{\mathbf{A} \sim \pi_\theta(\mathbf{Q})} \left[ R_{\text{RLVR}}(\mathbf{Q}, \mathbf{A}) \right] = \left[ R(\mathbf{Q}, \mathbf{A}) - \beta \text{KL} \left[ \pi_\theta(\mathbf{A} \mid \mathbf{Q}) \,\|\, \pi_{\text{ref}}(\mathbf{A} \mid \mathbf{Q}) \right] \right], \tag{6}$$

where $\pi_\theta$ and $\pi_{\text{ref}}$ are the policy model and the reference model, respectively. $R$ is the verifiable reward function. $\text{KL} \left[ \pi_\theta(\mathbf{A} \mid \mathbf{Q}) \,\|\, \pi_{\text{ref}}(\mathbf{A} \mid \mathbf{Q}) \right]$ penalizes divergence from the reference policy $\pi_{\text{ref}}$, ensuring both correctness and alignment with prior knowledge. The hyperparameter $\beta$ controls the trade-off between reward maximization and policy regularization.

GRPO directly compares the relative quality of responses within a group without requiring an additional critic model. Specifically, given a question $\mathbf{Q}$, GRPO first generates $G$ candidate responses $\{o_1, o_2, \ldots, o_G\}$ according to the current policy $\pi_{\theta_{\text{old}}}$, which are then assigned rewards $\{r_1, r_2, \ldots, r_G\}$. The relative quality of these responses is calculated by normalizing the rewards using their mean and standard deviation. GRPO encourages the model to prioritize responses with higher relative rewards, fostering improved performance without requiring a separate critic.

The verifiable reward function $R$ consists of the Format Reward and the Jaccard Reward, *i.e.*, $R = R_{\text{F}} + R_{\text{J}}$, ensuring both prediction accuracy and structural consistency. In line with DeepSeek-R1, the Format Reward $R_{\text{F}}$ is used to enforce the model's compliance with predefined formatting rules for the `<think>` and `<answer>` tags. The Jaccard Reward $R_J$ is a novel, meticulously designed reward function tailored for multi-disease diagnosis, which evaluates the alignment between the model's predictions and the ground truth by leveraging the Jaccard similarity coefficient, thereby quantifying the overlap between the predicted and actual disease sets. Specifically, for each model completion and its corresponding ground truth, the disease sets within the `<answer>` tags are first extracted using regular expressions and denoted as $L_C = \{l_{c_1}, l_{c_2}, \ldots, l_{c_m}\}$ and $L_G = \{l_{g_1}, l_{g_2}, \ldots, l_{g_n}\}$,

| Method | LLM | NLG | | | | CE | | | |
|---|---|---|---|---|---|---|---|---|---|
| | | BLEU | METEOR | ROUGE | BERT | PRECISION | RECALL | F1 SCORE | AUC |
| *General-purpose MLLMs* | | | | | | | | | |
| InternVL3-1B (Zhu et al., 2025) | InternVL3-1B | 0.0178 | 0.1884 | 0.1265 | 0.8188 | 0.3333 | 0.1333 | 0.1904 | 0.5053 |
| LLaVA-1.5-7B (Liu et al., 2024b) | Vicuna-7B | 0.0029 | 0.0809 | 0.0681 | 0.7796 | 0.2495 | 0.1279 | 0.1691 | 0.5004 |
| LLaVA-One-Vision-7B (Li et al., 2024b) | Qwen2-7B | 0.0144 | 0.1618 | 0.1168 | 0.8016 | 0.3120 | 0.1247 | 0.1782 | 0.4975 |
| Qwen2.5-VL-3B-Instruct (Bai et al., 2025) | Qwen2.5-3B-Instruct | 0.0218 | 0.2031 | 0.1331 | 0.8181 | 0.3493 | 0.1397 | 0.1995 | 0.5000 |
| Mini-InternVL-Chat-2B-V1-5 (Bai et al., 2025) | InternLM2-Chat-1.8B | 0.0092 | 0.1347 | 0.0959 | 0.8008 | 0.2176 | 0.1343 | 0.1661 | 0.5015 |
| Molmo-7B-O-0924 (Deitke et al., 2024) | OLMo-7B | 0.0155 | 0.1456 | 0.1070 | 0.8028 | 0.0295 | 0.0608 | 0.0398 | 0.5001 |
| Deepseek-VL-1.3B-Chat (Lu et al., 2024a) | Deepseek-1.3B-Chat | 0.0341 | 0.1756 | 0.1435 | 0.8128 | 0.2510 | 0.1278 | 0.1694 | 0.5021 |
| LLaVA-NeXT-8B (Li et al., 2024a) | LLaMA3-8B | 0.0145 | 0.1532 | 0.1067 | 0.8145 | 0.2674 | 0.1294 | 0.1744 | 0.4987 |
| *Medical domain-specific MLLMs* | | | | | | | | | |
| Med-Flamingo (Moor et al., 2023) | LLaMA2-7B | 0.0567 | 0.2134 | 0.1568 | 0.8328 | 0.3255 | 0.1427 | 0.1984 | 0.5201 |
| LLaVA-Med (Li et al., 2023b) | LLaVA-7B | 0.0735 | 0.2358 | 0.1637 | 0.8321 | 0.3028 | 0.1578 | 0.2075 | 0.5318 |
| HuatuoGPT-Vision (Chen et al., 2024a) | LLaVA-v1.6-34B | 0.0624 | 0.2017 | 0.1389 | 0.8048 | 0.2867 | 0.1622 | 0.2072 | 0.5038 |
| MedTVT-R1 w/o PT | LLaMA3.2-1B | 0.1131 | 0.3280 | 0.2043 | 0.8599 | 0.4980 | 0.5208 | 0.4672 | 0.5851 |
| MedTVT-R1 w/o RFT | LLaMA3.2-1B | 0.1325 | 0.3499 | 0.2261 | 0.8660 | 0.5237 | 0.5783 | 0.4992 | 0.6242 |
| **MedTVT-R1** | LLaMA3.2-1B | **0.1353** | **0.3536** | **0.2295** | **0.8652** | **0.5407** | **0.5908** | **0.5190** | **0.6554** |

Table 1: Comparison of MedTVT-R1 with various MLLMs and its variants on disease-level reasoning and diagnostic capabilities. The table is divided into general-purpose MLLMs and medical domain-specific MLLMs. The proposed MedTVT-R1 highlighted in gray.

where $l_{c_i}$ and $l_{g_j}$ represent individual diseases in the predicted and ground truth sets, respectively. The Jaccard Reward $R_J$ is then computed as:

$$R_J(L_C, L_G) = \begin{cases} \frac{|L_C \cap L_G|}{|L_C \cup L_G|}, & \text{if } |L_C \cup L_G| > 0, \\ 0, & \text{if } |L_C \cup L_G| = 0. \end{cases} \quad (7)$$

When the union of the sets is not empty, the $R_J$ is determined by the ratio of the intersection size to the union size, thereby capturing the degree of overlap between the prediction and ground truth. If the union is empty, the $R_J$ is set to zero to ensure robustness against invalid or incomplete outputs. Therefore, the Jaccard reward encourages the model to generate outputs that are highly consistent with the ground truth labels, which effectively helps improve both the accuracy and reliability in multi-disease diagnosis scenarios.

## 4 EXPERIMENTS

### 4.1 TRAINING DETAILS AND METRICS

**Training Details.** We conduct all experiments on a server equipped with eight NVIDIA A800 80GB GPUs. For the LLM, we choose LLaMA 3.2-1B (Grattafiori et al., 2024) and integrate the LoRA modules (Hu et al., 2022) with a rank of 8 for fine-tuning. For the modality-specific encoders, we use the pre-trained weights from ECGFM-KED (Tian et al., 2024), ViT-B/16 (Dosovitskiy et al., 2020), and Symile (Saporta et al., 2024) for ECG, CXR, and LAB, respectively. All modality-specific projectors adopt the Dense block architecture from MuMu-LLaMA (Liu et al., 2024c), with the embedding dimension $d$ set to 2048. During training, the PT and SFT stages are each trained for 20 epochs, while the RFT stage is trained for 500 iterations using the open-source Trainer framework, with $G$ in GRPO set to 8. **Metrics.** The effectiveness of multi-disease reasoning and diagnosis was evaluated from two perspectives. First, the descriptive accuracy of the generated diagnostic text was assessed using natural language generation (NLG) metrics, including BLEU, METEOR, ROUGE, and BERTScore. Second, the classification accuracy of multi-label disease categories in the responses was evaluated using clinical efficacy (CE) metrics, such as PRECISION, RECALL, F1 SCORE, and AUC.

### 4.2 QUANTITATIVE ANALYSIS

**Disease-level Diagnostic Reasoning Results.** Since no existing multimodal large model can jointly process ECG signals, medical images, and tabular data, we convert ECG signals into images and LAB data into text for a fair comparison. Table 1 reports results for our MedTVT-R1 against two categories of state-of-the-art MLLMs: *(i) General-purpose* — InternVL3-1B (Zhu et al., 2025), LLaVA-1.5-7B (Liu et al., 2024b), LLaVA-One-Vision-7B (Li et al., 2024b), Qwen2.5-VL-3B-Instruct (Bai et al., 2025), Mini-InternVL-Chat-2B-V1-5 (Bai et al., 2025), Molmo-7B-O-0924 (Deitke et al., 2024), Deepseek-VL-1.3B-Chat (Lu et al., 2024a), and LLaVA-NeXT-8B (Li et al., 2024a); *(ii) Medical-specific* — Med-Flamingo (Moor et al., 2023), LLaVA-Med (Li et al., 2023b), and HuatuoGPT-Vision (Chen et al., 2024a). These models span 1B–8B parameters and various backbones (InternVL, Vicuna, Qwen, OLMo, Deepseek, LLaMA3), with some incorporating RL or domain-specific pre-training. All inference results were obtained using ModelScope SWIFT (Zhao et al., 2025).

| Method | ECG-QA | | | | CXR-QA | | | | LAB-QA | | | |
|---|---|---|---|---|---|---|---|---|---|---|---|---|
| | BLEU | METEOR | ROUGE | BERT | BLEU | METEOR | ROUGE | BERT | BLEU | METEOR | ROUGE | BERT |
| *General-purpose MLLMs* | | | | | | | | | | | | |
| InternVL3-1B (Zhu et al., 2025) | 0.0186 | 0.1795 | 0.1379 | 0.8282 | 0.0239 | 0.1827 | 0.1273 | 0.8309 | 0.0083 | 0.1234 | 0.0750 | 0.7750 |
| LLaVA-1.5-7B (Liu et al., 2024b) | 0.0055 | 0.1084 | 0.0866 | 0.8100 | 0.0034 | 0.0967 | 0.0812 | 0.8012 | 0.0170 | 0.1402 | 0.1133 | 0.7937 |
| LLaVA-One-Vision-7B (Li et al., 2024b) | 0.0313 | 0.2263 | 0.1545 | 0.8322 | 0.0260 | 0.1877 | 0.1325 | 0.8214 | 0.0088 | 0.1362 | 0.0967 | 0.7883 |
| Qwen2.5-VL-3B-Instruct (Bai et al., 2025) | 0.0304 | 0.2483 | 0.1687 | 0.8418 | 0.0310 | 0.1798 | 0.1261 | 0.8230 | 0.0081 | 0.1129 | 0.0764 | 0.7832 |
| Mini-InternVL-Chat-2B-V1-5 (Bai et al., 2025) | 0.0102 | 0.1336 | 0.0984 | 0.8112 | 0.0088 | 0.1082 | 0.0825 | 0.8044 | 0.0085 | 0.1286 | 0.1118 | 0.7781 |
| Molmo-7B-O-0924 (Deitke et al., 2024) | 0.0233 | 0.1949 | 0.1341 | 0.8305 | 0.0211 | 0.1813 | 0.1255 | 0.8231 | 0.0091 | 0.1102 | 0.1120 | 0.7587 |
| Deepseek-VL-1.3B-Chat (Lu et al., 2024a) | 0.0240 | 0.1708 | 0.1162 | 0.8205 | 0.0298 | 0.1510 | 0.1312 | 0.8215 | 0.0118 | 0.0975 | 0.1184 | 0.7675 |
| LLaVA-NeXT-8B (Li et al., 2024a) | 0.0091 | 0.1412 | 0.1064 | 0.8009 | 0.0107 | 0.1305 | 0.1307 | 0.8199 | 0.0102 | 0.1057 | 0.1091 | 0.7623 |
| *Medical domain-specific MLLMs* | | | | | | | | | | | | |
| Med-Flamingo (Moor et al., 2023) | 0.0526 | 0.2016 | 0.1728 | 0.8517 | 0.0527 | 0.1978 | 0.1567 | 0.8326 | 0.0190 | 0.1178 | 0.1236 | 0.7768 |
| LLaVA-Med (Li et al., 2023b) | 0.0728 | 0.2218 | 0.1829 | 0.8328 | 0.0618 | 0.2119 | 0.1418 | 0.8418 | 0.0256 | 0.1327 | 0.1318 | 0.7826 |
| HuatuoGPT-Vision (Chen et al., 2024a) | 0.0758 | 0.2518 | 0.1910 | 0.8529 | 0.0719 | 0.2249 | 0.1528 | 0.8518 | 0.0211 | 0.1411 | 0.1419 | 0.7736 |
| **MedTVT-R1** | **0.0831** | **0.3044** | **0.2202** | **0.8650** | **0.0931** | **0.3073** | **0.2121** | **0.8673** | **0.1807** | **0.3827** | **0.3081** | **0.8855** |

Table 2: Comparison of MedTVT-R1 with various MLLMs on physiological-level understanding and analysis capabilities. The table is divided into general-purpose MLLMs and medical domain-specific MLLMs. The proposed MedTVT-R1 highlighted in gray.

MedTVT-R1 surpasses both general-purpose and medical-specific baselines in natural language generation and clinical evaluation, demonstrating superior descriptive and diagnostic reasoning for multi-disease scenarios. Table 1 also includes ablations: removing physiological-level pre-training or GRPO-based RFT post-training degrades performance, confirming that 1) physiological-level pre-training endows the model with cross-modal physiological representation capabilities, facilitating more effective multimodal integration during SFT; and 2) GRPO-based RFT leverages the available training data to refine and strengthen multi-disease diagnostic reasoning.

**Physiological-level Understanding Results.** Following the setup in Disease-level Diagnostic Reasoning, we evaluated MedTVT-R1 on single-modality physiological-level understanding against two categories of state-of-the-art MLLMs. Results in Table 2 show MedTVT-R1 outperforms all competitors. Notably, the designed Physiological-level representation analysis is a challenging long-text generation task ($\geq 300$ words per instance); MedTVT-R1 still delivers outstanding results, highlighting its strength in handling lengthy, detailed outputs. These findings confirm MedTVT-R1's exceptional performance in long-text generation, effective comprehension of cross-modal physiological data, and robustness in both single-modality perception and multimodal reasoning, establishing it as a leading solution for complex medical data analysis.

**Synergistic Advantage of Multi-Modal Integration.** To quantify the performance gains derived from the joint utilization of ECG, CXR, and laboratory test data in multi-disease diagnosis, we conducted a modality ablation study. The model was evaluated under seven input configurations: (1) all three modalities, (2–4) pairwise modality combinations, and (5–7) single-modality inputs. For absent modalities, the corresponding input tensor was substituted with zero-valued tensors of matching dimensionality to preserve architectural consistency and isolate the impact of modality exclusion. As reported in Table 3, the full tri-modal configuration attained

Table 3: Performance comparison under different modality combinations. The values in parentheses indicate the relative performance drop (%) compared to the full three-modality setting.

| Modalities | Micro F1 | Macro F1 | Jaccard |
|---|---|---|---|
| Full | 0.519 | 0.457 | 0.389 |
| NoLab | 0.488 (-6.0%) | 0.374 (-18.2%) | 0.352 (-9.5%) |
| NoCXR | 0.482 (-7.1%) | 0.409 (-10.5%) | 0.320 (-17.7%) |
| NoECG | 0.484 (-6.7%) | 0.415 (-9.2%) | 0.322 (-17.2%) |
| ECGOnly | 0.460 (-11.4%) | 0.351 (-23.2%) | 0.327 (-15.9%) |
| CXROnly | 0.470 (-9.4%) | 0.353 (-22.7%) | 0.333 (-14.4%) |
| LabOnly | 0.482 (-7.1%) | 0.413 (-9.6%) | 0.328 (-15.7%) |

the highest Micro F1 (0.519), Macro F1 (0.457), and Jaccard (0.389) scores. Excluding any modality led to measurable performance degradation, with the most pronounced declines in Macro F1 observed when CXR or laboratory data were omitted—indicating that specific disease categories derive substantial discriminative benefit from particular modalities. Single-modality inputs yielded markedly inferior results, especially in Macro F1, evidencing the complementary representational strengths of ECG, CXR, and laboratory data. Collectively, these findings substantiate that tri-modal fusion confers a synergistic representational advantage, facilitating more balanced and resilient diagnostic performance across heterogeneous disease categories.

**Effect of MPL Design and Modality Completeness.** Table 4 ablation results confirm the effectiveness of Cyclic Multi-Head Attention (CMHA) and Contribution-Aware Operator (CAO) in the Modality Perception Layer (MPL), plus the criticality of full modality integration in pre-training. CMHA/CAO incorporation boosts performance, validating designs for enhanced modality fusion and adaptive modality weighting across diseases. For modality composition, fusing all three modalities (ECG, CXR, LAB) yields the best results; removing any degrades performance, with ECG exclusion causing the sharpest METEOR/ROUGE drops (attributed to strong disease-cardiac activity associations). These findings emphasize that advanced fusion mechanisms and full modality integration are both essential for optimal diagnostic reasoning.

Table 4: The ablation of the Cyclic Multi-Head Attention (CMHA) and Contribution-Aware Operator (CAO) in the Modality Perception Layer (MPL), and the impact of modal missingness during pre-training. Metrics: METEOR, ROUGE, RECALL, F1. Default settings are highlighted.

(a) Ablation study of MPL.

| MPL | | NLG | | CE | |
|-----|-----|-----|-----|-----|-----|
| CMHA | CAO | MET. | ROU. | REC. | F1 |
| ✗ | ✓ | 0.3455 | 0.2013 | 0.5733 | 0.4977 |
| ✓ | ✗ | 0.3378 | 0.2145 | 0.5826 | 0.4867 |
| ✓ | ✓ | 0.3536 | 0.2295 | 0.5908 | 0.5190 |

(b) Ablation study of modalities.

| Modality | | | NLG | | CE | |
|-----|-----|-----|-----|-----|-----|-----|
| ECG | CXR | LAB | MET. | ROU. | REC. | F1 |
| ✗ | ✓ | ✓ | 0.3245 | 0.2058 | 0.5320 | 0.4739 |
| ✓ | ✗ | ✓ | 0.3267 | 0.2019 | 0.5739 | 0.4869 |
| ✓ | ✓ | ✗ | 0.3455 | 0.2218 | 0.5845 | 0.5023 |
| ✓ | ✓ | ✓ | 0.3536 | 0.2295 | 0.5908 | 0.5190 |

Figure 4: Inference examples of MedTVT-R1. Left: successful case with accurate multimodal diagnosis; Right: failure case where coexisting cardiac, renal, and metabolic abnormalities led the model to prioritize atrial fibrillation and miss sepsis.

## 4.3 QUALITATIVE ANALYSIS

MedTVT-R1 integrates chest X-rays (CXR), electrocardiograms (ECG), and laboratory (LAB) data into coherent, evidence-supported diagnostic conclusions (Figure 4). We illustrate its behavior through one high-performing case and one complex multi-disease case from two perspectives: *Multimodal Integration* and *Evidence-Based Reasoning*. **1) Multimodal Integration.** In a high-performing case (*Coronary Artery Disease*, *Hypertension*), complementary ECG, CXR, and LAB findings align to reinforce diagnoses, with cross-modal evidence yielding precise and clinically consistent results. The right panel illustrates a complex failure case under interference from multiple coexisting diseases: when cardiac, renal, and metabolic abnormalities occur simultaneously, the model tends to prioritize high-confidence conditions and overlook acute or subtle ones. Here, ECG findings of "irregular rhythm" and absent P waves led to an atrial fibrillation diagnosis, resulting in a missed sepsis diagnosis. Such situations, arising from limited supporting modality evidence, may benefit from incorporating additional modalities to strengthen the evidence chain. **2) Evidence-Based Reasoning.** In the high-performing case, the model explicitly links features across modalities—e.g., ECG-derived left ventricular hypertrophy with CXR cardiac enlargement—to justify conclusions. In the complex case, reasoning is dominated by strong single-modality cues, underscoring the potential of expanded multimodal inputs to improve cross-modal reasoning consistency. These examples highlight MedTVT-R1's strength in synthesizing heterogeneous clinical evidence, while complex scenarios reveal opportunities to enhance diagnostic coverage through richer multimodal evidence.

## 5 CONCLUSION

In conclusion, the proposed MedTVT-R1 framework represents a significant advancement in the application of multimodal large language models (MLLMs) for medical diagnosis. By integrating the complementary strengths of ECG, CXR, and LAB data, MedTVT-R1 addresses the limitations of single-modal approaches and provides a more holistic understanding of complex diseases. The innovative MedTVT-QA dataset facilitates physiological perception and multi-disease diagnosis by leveraging a Chain of Evidence strategy. Additionally, the modality perception layer enhances cross-modal interactions, while Reinforcement Fine-Tuning with Group Relative Policy Optimization and the Jaccard Reward boosts precision and reliability in diagnosis capabilities. Extensive experiments validate MedTVT-R1's superior performance in both physiological-level understanding and disease-level diagnosis, highlighting its potential for practical clinical applications, such as interpretable diagnostic report generation and complex comorbidity reasoning.

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

# APPENDIX

## A  DETAILS ABOUT THE PROMPTS OF MEDTVT-QA'S CONSTRUCTION

This section presents the detailed prompts used in constructing the MedTVT-QA dataset.

---

**ECG-QA Prompt**

***Role Setting***: You are a renowned cardiologist with expertise in interpreting electrocardiograms (ECGs).
***Task Description***: The ECG analysis has yielded the following labels: {labels}. Based on these labels, please address the question: {question}.
***Answer Guidance***: Your response should incorporate all relevant labels, excluding any unrelated ones. Provide a synthesis of the labels, focusing on clinical significance.
***Answer Format***: Begin with a brief introduction to your analysis. Provide detailed explanations for each specific ECG label. Offer a concise summary.

---

**CXR-QA Prompt**

***Role Setting***: You are a radiology expert with expertise in interpreting chest X-ray image.
***Task Description***: The chest X-ray report is given {report} Base on the given chest X-ray report, answer the question {question}
***Answer Guidance***: Describe the overall condition of the lungs, heart, and chest cavity in the image. Identify and explain any abnormal findings such as shadows, opacities, effusions, or masses. Provide possible diagnoses.

---

**LAB-QA Prompt**

***Role Setting***: Please analyze this set of blood test data as a medical professional.
***Task Description***: This is the question: {question} The following are the lab data: "Hematocrit": {data[0]}; "Platelet Count": {data[1]}; "Creatinine": {data[2]}; "Potassium": {data[3]}; "Hemoglobin": {data[4]}; "White Blood Cells": {data[5]}; "MCHC": {data[6]}; "Red Blood Cells": {data[7]}; "MCV": {data[8]}; "MCH": {data[9]}; "RDW": {data[10]}; "Urea Nitrogen": {data[11]}; "Sodium": {data[12]}; "Chloride": {data[13]}; "Bicarbonate": {data[14]}; "Anion Gap": {data[15]}; "Glucose": {data[16]}; "Magnesium": {data[17]}; "Calcium, Total": {data[18]}; "Phosphate": {data[19]}; "INR(PT)": {data[20]}; "PT": {data[21]}; "PTT": {data[22]}; "Basophils": {data[23]}; "Neutrophils": {data[24]}; "Monocytes": {data[25]}; "Eosinophils": {data[26]}; "Lymphocytes": {data[27]}; "RDW-SD": {data[28]}; "H": {data[29]}; "L": {data[30]}; "I": {data[31]}; "Alanine Aminotransferase (ALT)": {data[32]}; "Asparate Aminotransferase (AST)": {data[33]}; "Lactate": {data[34]}; "Alkaline Phosphatase": {data[35]}; "Bilirubin, Total": {data[36]}; "pH": {data[37]}; "Albumin": {data[38]}; "Base Excess": {data[39]}; "pO2": {data[40]} "Calculated Total CO2": {data[41]}; "pCO2": {data[42]}; "Absolute Neutrophil Count": {data[43]}; "Absolute Eosinophil Count": {data[44]}; "Absolute Monocyte Count": {data[45]}; "Absolute Basophil Count": {data[46]}; "Absolute Lymphocyte Count": {data[47]}; "Creatine Kinase (CK)": {data[48]} "Immature Granulocytes": {data[49]}
***Answer Guidance***: These data comprise 50 different indicators, categorized into seven main classes: routine blood indicators, electrolyte and metabolic indicators, renal function indicators, liver function indicators, acid-base balance and gas exchange, coagulation function indicators, and other indicators.
***Answer Format***: Begin with a brief introduction to your analysis.
**routine blood indicators**: explanation
**electrolyte and metabolic indicators**: explanation
**renal function indicators**: explanation
**liver function indicators**: explanation
**acid-base balance and gas exchange**: explanation
**coagulation function indicators**: explanation
**other indicators**: explanation
Finally, offer a concise summary.

---

---

**Disease-QA Prompt**

*Role Setting*: You are a renowned diagnostician with expertise in integrating ECG, CXR, and blood test results.
*Task Description*: The following diagnostics have been provided:
• ECG Analysis: {ecg_report}
• CXR Analysis: {cxr_report}
• Blood Test Analysis: {blood_test_report}
• Diseases: {result_diseases}
You need to pretend that the ECG, CXR, and blood test analyses are based on your interpretation of the raw data, and the final diagnosis is your synthesis of these three diagnostic methods, please address the question: {question}
*Answer Guidance*: Please find definitive evidence from the ECG, CXR, and blood test results, leveraging the complementarity and mutual corroboration of these three modalities, to robustly prove the reasons why the patient has the diseases I provided. Your response must include every disease I provided, using the exact wording I provided, and you must not mention any diseases other than those I provided. Please make sure to provide evidence for these diagnoses! These are confirmed conditions.
*Answer Format*: `<think>`{Diagnostic evidence synthesized from the three modalities}`</think>`\n `<answer>`{disease1; disease2; ...}`</answer>`

---

# B    LABEL DISTRIBUTION OF MEDTVT-QA

When constructing the physiology-level ECG-QA dataset, we filtered out invalid ECG labels to ensure that the final labels align with morphology descriptions at the physiological level. Additionally, we conducted a detailed statistical analysis of the labels in the ECG-QA training data. As shown in Table 5, it presents ECG labels with occurrences greater than 100 along with their respective counts.

Table 5: ECG Labels and Counts ($\dot{\iota}$100) in ECG-QA.

| Label | Count |
| --- | --- |
| sinus rhythm with 1st degree a-v block | 140 |
| sinus rhythm | 4033 |
| atrial fibrillation | 761 |
| sinus tachycardia | 1565 |
| consider acute st elevation mi | 161 |
| atrial fibrillation with rapid ventricular response | 224 |
| age not entered, assumed to be 50 years old for purpose of ecg interpretation | 328 |
| sinus bradycardia | 402 |
| sinus rhythm with pac(s) | 132 |
| sinus rhythm with borderline 1st degree a-v block | 121 |
| pacemaker rhythm - no further analysis | 160 |
| leftward axis | 435 |
| possible left anterior fascicular block | 138 |
| rightward axis | 164 |
| probable left atrial enlargement | 224 |
| low qrs voltages in precordial leads | 540 |
| st junctional depression is nonspecific | 149 |
| possible inferior infarct - age undetermined | 425 |
| lateral t wave changes are nonspecific | 328 |
| short pr interval | 167 |
| inferior t wave changes are nonspecific | 312 |
| left ventricular hypertrophy | 428 |
| lvh with secondary repolarization abnormality | 285 |
| left axis deviation | 1067 |
| poor r wave progression - probable normal variant | 538 |
| indeterminate axis | 108 |
| possible anterior infarct - age undetermined | 511 |
| anterior t wave changes are nonspecific | 182 |
| possible left atrial abnormality | 271 |
| inferior/lateral st-t changes are nonspecific | 240 |
| prolonged qt interval | 618 |
| possible anteroseptal infarct - age undetermined | 254 |
| septal t wave changes are nonspecific | 134 |
| right bundle branch block | 517 |
| lateral st-t changes are nonspecific | 289 |
| anteroseptal infarct - age undetermined | 129 |
| left anterior fascicular block | 202 |

| Label | Count |
|---|---|
| extensive st-t changes are nonspecific | 111 |
| inferior infarct - age undetermined | 550 |
| rsr'(v1) - probable normal variant | 199 |
| left bundle branch block | 354 |
| low qrs voltages in limb leads | 395 |
| extensive st-t changes may be due to myocardial ischemia | 143 |
| possible left ventricular hypertrophy | 150 |
| abnormal r-wave progression, early transition | 102 |
| inferior infarct, old | 123 |
| ventricular premature complex | 119 |
| possible septal infarct - age undetermined | 188 |
| right axis deviation | 141 |
| lateral st-t changes may be due to myocardial ischemia | 227 |
| inferior/lateral st-t changes may be due to myocardial ischemia | 167 |
| iv conduction defect | 376 |
| generalized low qrs voltages | 161 |
| qrs changes v3/v4 may be due to lvh but cannot rule out anterior infarct | 103 |
| lateral t wave changes may be due to myocardial ischemia | 106 |
| rbbb with left anterior fascicular block | 314 |
| extensive st-t changes may be due to hypertrophy and/or ischemia | 135 |
| normal ecg | 753 |
| normal ecg except for rate | 334 |
| abnormal ecg | 4761 |
| borderline ecg | 2074 |
| inferior/lateral st-t changes may be due to hypertrophy and/or ischemia | 116 |
| lateral st-t changes may be due to hypertrophy and/or ischemia | 112 |

Figure 5 presents an example report from the MIMIC-IV-CXR dataset, used in constructing CXR-QA. The report contains some unclear and unrelated content to the CXR image description. By applying the previously described CXR prompts, we transformed the report into a more organized and focused description centered on CXR.

Figure 5: An CXR report example from MIMIX-IV-CXR-report dataset.

**FINAL REPORT**

**EXAMINATION:** CHEST (PA AND LAT)
**INDICATION:** __F with new onset asciteps // eval for infection

**TECHNIQUE:** Chest PA and lateral

**COMPARISON:** None.

**FINDINGS:** There is no focal consolidation, pleural effusion or pneumothorax. Bilateral nodular opacities that most likely represent nipple shadows.

The cardiomediastinal silhouette is normal. Clips project over the left lung, potentially within the breast. The imaged upper abdomen is unremarkable. Chronic deformity of the posterior left sixth and seventh ribs are noted.
**IMPRESSION:** No acute cardiopulmonary process.

Disease-level labels are derived from the MIMIC-IV-ECG-EXT-ICD (Strodthoff et al., 2024) dataset, with these labels stored as ICD-10 codes. Each sample may correspond to multiple disease categories. We filtered out diseases for which evidence could not be found in ECG, CXR, or LAB data. Ultimately, we identified seven main categories: Coronary Artery Disease, Acute Renal Failure, Hypertension, Atrial Fibrillation, Pneumonia, Diabetes Mellitus, and Sepsis, along with some subclasses within these categories. Details are provided in Table 6.

Table 6: ICD-10 Disease Statistics with Corresponding Counts.

| Disease Category | ICD-10 Code | Count |
|---|---|---|
| **Coronary Artery Disease** | | |
| Coronary Artery Disease | I2510 | 2680 |
| Chronic ischemic heart disease, unspecified | I252 | 936 |
| Atherosclerotic heart disease of native coronary artery | I259 | 190 |
| Other forms of chronic ischemic heart disease | I253 | 8 |
| Ischemic cardiomyopathy | I255 | 79 |
| **Acute Renal Failure** | | |
| Acute kidney failure, unspecified | N179 | 2379 |
| Acute kidney failure with tubular necrosis | N170 | 689 |

| Disease Category | ICD-10 Code | Count |
|---|---|---|
| Acute kidney failure with other specified morphologic lesions | N178 | 12 |
| Acute kidney failure with acute cortical necrosis | N171 | 1 |
| **Hypertension** | | |
| Essential (primary) hypertension | I10 | 4155 |
| Hypertensive heart and chronic kidney disease, unspecified | I129 | 1536 |
| Hypertensive heart disease with heart failure | I120 | 515 |
| Hypertensive heart and chronic kidney disease with heart failure | I130 | 119 |
| Hypertensive heart disease without heart failure | I110 | 77 |
| Hypertensive heart and chronic kidney disease with heart failure and stage 5 CKD or ESRD | I132 | 37 |
| Hypertensive heart disease, unspecified | I119 | 8 |
| Other secondary hypertension | I159 | 1 |
| Renovascular hypertension | I150 | 7 |
| Other specified secondary hypertension | I158 | 1 |
| **Atrial Fibrillation** | | |
| Persistent atrial fibrillation | I4891 | 2623 |
| Permanent atrial fibrillation | I4892 | 317 |
| Paroxysmal atrial fibrillation | I480 | 237 |
| Other specified atrial fibrillation | I482 | 104 |
| Atrial flutter, unspecified | I481 | 15 |
| Typical atrial flutter | I483 | 1 |
| Atypical atrial flutter | I484 | 1 |
| **Pneumonia** | | |
| Pneumonia, unspecified organism | J189 | 1442 |
| Pneumonia due to other specified bacteria | J181 | 25 |
| Pneumonia due to Haemophilus influenzae | J188 | 18 |
| Pneumonia due to Klebsiella pneumoniae | J180 | 5 |
| **Diabetes Mellitus** | | |
| Type 2 diabetes mellitus without complications | E119 | 1752 |
| Type 2 diabetes mellitus with unspecified complications | E1129 | 217 |
| Type 2 diabetes mellitus with hyperglycemia | E11319 | 296 |
| Type 2 diabetes mellitus with renal complications | E1140 | 348 |
| Type 2 diabetes mellitus with circulatory complications | E1165 | 360 |
| Type 2 diabetes mellitus with other specified complications | E118 | 29 |
| Type 2 diabetes mellitus with unspecified complications | E139 | 29 |
| Type 1 diabetes mellitus without complications | E109 | 27 |
| Type 1 diabetes mellitus with unspecified complications | E138 | 2 |
| Type 1 diabetes mellitus with other specified complications | E108 | 3 |
| **Sepsis** | | |
| Sepsis, unspecified organism | A419 | 1088 |
| Severe sepsis without septic shock | R6520 | 939 |
| Severe sepsis with septic shock | R6521 | 873 |
| Sepsis due to Streptococcus pneumoniae | A403 | 26 |
| Sepsis due to Escherichia coli | A412 | 2 |
| Sepsis due to other Gram-negative organisms | A409 | 69 |
| Sepsis due to anaerobes | A414 | 28 |
| Sepsis due to other specified bacteria | A411 | 26 |
| Sepsis due to Staphylococcus aureus | A401 | 3 |
| Sepsis due to other specified Staphylococcus | A408 | 4 |
| Sepsis due to other specified bacteria | A413 | 1 |
| Sepsis due to unspecified Staphylococcus | A400 | 2 |

# C  EXAMPLES OF QA PAIRS CORRESPONDING TO EACH MODALITY

This section provides examples of question-answer pairs corresponding to each modality (e.g., ECG, CXR, LAB) in the MedTVT-QA dataset.

---

**ECG-QA Example**

***Question***: What information can be derived from the ECG ¡ecg¿?

***Answer***: ECG Analysis: Upon evaluating the electrocardiogram (ECG), the following observations can be made:

1.Sinus Rhythm: The heart is under the control of the sinoatrial (SA) node, which is the natural pacemaker. This finding suggests that the electrical activity of the heart is originating from the SA node and progressing through the normal conduction pathway, leading to a regular rhythm. This is a foundational indicator of organized cardiac activity.

2. Left Bundle Branch Block (LBBB): There is evidence of delayed or disrupted conduction through the left bundle of the heart's electrical conduction system. This results in a characteristic pattern on the ECG, with a widened QRS complex (¿120 ms) and specific morphologies in leads, such as a broad, notched appearance in V5-V6 and an absence of normal septal depolarization. LBBB may occur in conditions like ischemic heart disease, cardiomyopathy, or hypertension-induced structural changes, though it can also occasionally be found in asymptomatic individuals.

Summary: The heart demonstrates a sinus rhythm, indicating normal initiation of electrical activity. However, the presence of a left bundle branch block signifies an abnormality in electrical conduction, often associated with underlying cardiac pathology. As a whole, this ECG is classified as abnormal due to the conduction disturbance, warranting further investigation to assess structural or functional cardiac issues.

---

---

**CXR-QA Example**

*Question*: Would you mind explaining the findings on my chest X-ray ¡cxr¿?

*Answer*: Based on the evaluation of your chest X-ray image, the lungs appear overall unchanged from prior assessments, indicating no new substantial developments. There is no evidence of acute cardiopulmonary disease, which means there are no significant new concerns regarding your heart or lungs. Specifically, there are no signs of pneumonia, which would typically appear as clouded or consolidated areas in the lung fields. Similarly, there is no vascular congestion; this would generally be associated with heart-related problems affecting the lung's blood vessels and is absent in this image. Additionally, there is no pleural effusion, which would be fluid accumulation in the space between the lungs and chest wall.

However, there is a notable finding of a dense streak of atelectasis at the left base of the lung, which is a condition where a part of the lung collapses or does not inflate properly, resulting in that portion not being fully functional. This streak appears denser than usual tissue, indicating that a segment of the lung at the left base may not be fully expanded or aerated. Atelectasis can result from various causes, such as mucus blocking an airway, external pressure on the lung, or post-surgical changes.

Despite this abnormality, the absence of acute cardiopulmonary disease indicators is reassuring. The streak of atelectasis should be monitored, and clinical correlation with your symptoms is important. Continuous observation or additional imaging might be recommended to ensure there are no underlying issues contributing to this finding. Always consult with your healthcare provider to discuss the implications of these findings based on your clinical history and symptoms.

---

**LAB-QA Example**

*Question*: Would you be able to undertake a comprehensive examination of this patient's blood panel ¡lab¿, including the reference ranges and clinical implications associated with all abnormal values?

*Answer*: Comprehensive Blood Panel Analysis:This blood test reflects multiple systemic functions. I will analyze the provided values across seven categories, indicating normal findings and clinical implications for abnormal or missing data.

Routine Blood Indicators: Hematocrit, hemoglobin, red blood cells, MCV, and MCHC fall within normal ranges, suggesting typical oxygen-carrying capacity and erythrocyte health. RDW is borderline high, which could indicate subtle variability in red cell size, potentially pointing toward early nutritional deficiencies or mild anemia. Platelet count and white blood cells are normal, showing adequate clotting potential and immune activity. All differential leukocyte percentages are within normal ranges, reflecting balanced immunity.

Electrolyte and Metabolic Indicators: Potassium, sodium, magnesium, bicarbonate, and phosphate are within normal ranges, indicating balanced electrolytes and acid-base homeostasis. However, chloride is mildly low (97.0), potentially reflecting minor electrolyte imbalance or compensatory metabolic alkalosis. Glucose is significantly elevated (185.0), possibly suggesting poorly controlled diabetes mellitus or acute hyperglycemia. Lactate is elevated (4.6), which could suggest tissue hypoxia or metabolic distress. Calcium is borderline low (8.5), potentially indicating subtle hypocalcemia warranting further evaluation.

Renal Function Indicators: Creatinine is within the high-normal range, which might raise concerns for early renal insufficiency depending on age/gender. Urea nitrogen is elevated, possibly indicating dehydration, increased protein metabolism, or mild renal dysfunction.

Liver Function Indicators: ALT, AST, bilirubin, and alkaline phosphatase values are missing and assumed normal, precluding further analysis.

Acid-Base Balance and Gas Exchange: With missing pH and gas exchange values, bicarbonate is normal but anion gap is elevated (22.0), potentially indicating metabolic acidosis from lactate accumulation or unmeasured ions.

Coagulation Function Indicators: INR, PT, and PTT are missing and presumed normal, suggesting no overt coagulation abnormalities.

Other Indicators: Data gaps in albumin, base excess, and other parameters limit full metabolic insight.

Summary: Elevated glucose and lactate suggest metabolic dysregulation. Elevated urea nitrogen and borderline creatinine warrant renal assessment. Mild hypocalcemia and low chloride need further exploration. Elevated anion gap hints at underlying acidosis. Follow-up tests are recommended for confirmation and diagnosis refinement.

---

# D  FURTHER ANALYSIS

**Disease Co-occurrence Structure and Error Patterns.** As shown in Figure 6, the predicted co-occurrence matrix largely preserves the topology of the ground truth, recovering clinically plausible comorbidity clusters (e.g., hypertension–diabetes–coronary artery disease and hypertension–atrial fibrillation). Pairwise counts are modestly inflated in prediction, suggesting that the model leverages co-occurrence priors and tends to overproduce multi-label sets. The error profile in Figure 6 indicates systematic difficulty on lab-driven conditions. Diabetes mellitus and acute renal failure exhibit both high false positives and false negatives, pointing to unstable evidence due to missing labs, or heterogeneous diagnostic criteria. Sepsis also shows elevated false negatives, consistent with episodic onset and sparse specific biomarkers. In contrast, atrial fibrillation skews toward false negatives more than false positives, hinting at a conservative decision boundary or sensitivity to ECG noise/quality. Overall, the model learns a clinically meaningful dependency structure but occasionally substitutes co-occurrence priors for modality-specific evidence.

**Dynamic Modality Weights and Disease-level Performance.** To assess the behavior of the proposed dynamic modality-weighting mechanism, we visualized the learned weights for the three modalities using violin plots, as shown in Fig 7. The model assigns a broad range of weights across samples rather than converging to fixed or uniform values, indicating that reliance on each modality adapts on a per-case basis. The distributions are non-symmetric and, in some cases, multi-modal, suggesting the model can switch between modality-dominant decision modes depending on available evidence.

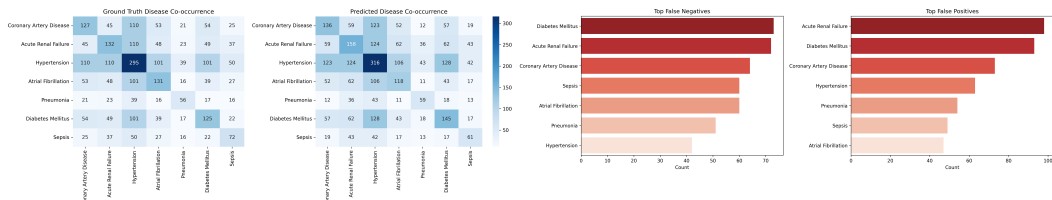

Figure 6: Disease co-occurrence and error profile. Left: Ground-truth vs. predicted co-occurrence matrices show similar structure with hypertension as a hub. Right: Top false negatives/positives highlight diabetes mellitus and acute renal failure as major sources of confusion.

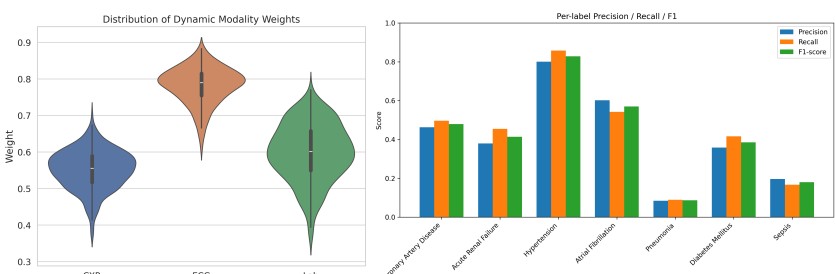

Figure 7: Dynamic modality weights and per-disease metrics. Left: Violin plots of learned weights for CXR, ECG, and Lab (median/IQR); ECG is highest and more stable, CXR lower, Lab most variable. Right: Per-label precision/recall/F1, showing alignment between higher ECG reliance and stronger cardiovascular performance, with pneumonia remaining weaker.

Linking weights to per-disease performance, the consistently higher ECG weights align with the stronger results on cardiovascular labels (e.g., hypertension and atrial fibrillation), reflecting the stable discriminative value of ECG for such conditions. CXR receives lower and less variable weights, which is consistent with the relatively weaker performance on pneumonia in our current setup—pointing to either under-exploitation of CXR features or label imbalance/noise. Lab shows the widest spread, consistent with its context-dependent value: it is more informative for organ-function–related conditions (e.g., acute renal failure) but contributes unevenly to others (e.g., sepsis, diabetes), likely due to missing tests and asynchrony.

## E   PERFORMANCE OF VARIOUS MLLMS IN HANDLING MEDICAL REASONING AND DIAGNOSIS

Here, we present an example of medical analysis and diagnosis among various MLLMs and MedTVT-R1, as shown in Figure 8.

## F   LIMITATION

Although our proposed MedTVT-R1 successfully integrates CXR, ECG, and LAB data for joint multi-disease diagnosis, there are still some limitations. First, precise disease diagnosis often requires a larger volume of multimodal data collected from the same patient within a similar timeframe. However, in the short term, it is challenging to gather such large-scale data, which limits the model's generalization ability and diagnostic accuracy. Second, while our work incorporates three modalities—CXR, ECG, and LAB—more accurate disease analysis and diagnosis may rely on additional modalities, such as patient medical history, genomic data, or other biomarkers. Unfortunately, the current open-source datasets lack richer multimodal information, making such extensions difficult to achieve. Future research could aim to address these challenges to further enhance the model's diagnostic capabilities and practical applicability.

What illnesses might be indicated by the findings from my ECG <ecg>, CXR <cxr>, and blood work <lab>?

**MedTVT-R1**

<think>Based on the ECG, CXR, and blood test analyses, **the findings** *collectively support* **the presence of Diabetes Mellitus and Hypertension**.
The diagnosis of **Diabetes Mellitus** is *substantiated* by blood test results. The markedly elevated glucose levels indicate hyperglycemia, a hallmark of diabetes. Although this blood test result suggests persistent elevated glucose levels, acute complications such as low glucose levels are characteristic of diabetes. Additionally, mild anemia and low albumin could reflect chronic disease-related nutritional deficits or underlying inflammatory processes often associated with diabetes. The low-normal albumin level may further suggest chronic systemic effects often seen in diabetes-related complications.
The diagnosis of **Hypertension** is *supported* by findings from all three diagnostic modalities. The ECG demonstrates a sinus rhythm with borderline findings, which can be associated with increased cardiac workload often seen in hypertensive patients. The presence of a small left-sided pleural effusion and mild pulmonary edema on the CXR aligns with fluid retention often seen in hypertensive heart disease or postoperative states. While the heart size is not detailed here, the pleural effusions seen on imaging may indirectly support hypertensive fluid shifts or cardiovascular strain. Elevated creatinine and urea nitrogen on the blood test suggest possible hypertensive nephropathy or reduced renal filtration rates. Hyperglycemia may further exacerbate vascular strain linked to hypertension.
In conclusion, *the evidence* points to Diabetes Mellitus through hyperglycemia, and Hypertension through cardiac strain and renal involvement clearly through indirect ECG and CXR findings. Both conditions require further management to mitigate systemic and cardiovascular risks.</think>
<answer>**Diabetes Mellitus; Hypertension**</answer>

**Qwen2.5-VL-3B-Instruct**

The provided ECG and chest X-ray images can be analyzed together to identify potential health issues. Here's a step-by-step analysis:
ECG Analysis:1. P Waves: The P waves appear normal in all leads (I, II, III, aVR, aVL, aVF, V1-V6). This suggests that the atrial activity is normal.
2. QRS Complexes: The QRS complexes are wide and have a duration of approximately 120ms , which is indicative of left ventricular hypertrophy (LVH).
3. T Waves: The T waves are not significantly abnormal in any lead.
4. ST Segments: The ST segments are within normal limits.
Chest X-Ray Analysis:
1. Cardiac Size: The cardiac silhouette appears enlarged, which could be due to LVH or other cardiomegaly causes.
2. Lung Fields: The lung fields appear clear with no signs of consolidation, effusion, or other abnormalities.
3. Heart Position: The heart position is normal, with no evidence of cardiomegaly or pericardial effusion.
Blood Test Results:
Creatinine: 0.2264026403427124 mg/dL (normal range: 0.6–1.2 mg/dL)
BUN: 0.2870349884033203 mg/dL (normal range: 7–20 mg/dL)
Calcium: 0.6595468521118164 mg/dL (normal range: 8.5–10.5 mg/dL)
Magnesium: 0.345366507768631 mg/dL (normal range: 1.7–2.6 mg/dL)
Potassium: 0.4424666166305542mEq /L (normal range: 3.5–5.0mEq/L)
Sodium: 0.524380087852478mEq /L (normal range: 135–145 mEq/L)
Chloride: 0.5399884581

**InternVL3-1B**

Based on the ECG and chest X-ray provided, the possible health concerns could include:
1. Arrhythmias: These include abnormal heart rhythms such as bradycardia, tachycardia, or atrial fibrillation, indicated by the presence of irregular QRS complexes.
2. Cardiomyopathy: Structural changes in the heart muscle, which might be visible on the ECG with abnormal heart rhythm patterns.
3. Valvular Aortic Stenosis: The semi-erect portable chest X-ray suggests possible pulmonary hypertension as the vascular structures seen suggest dilation in pulmonary valve and possibly aortic root shadow suggesting valve involvement.
4. Coronary Artery Disease: Though less likely in this chest X-ray alone, subtle changes could indicate coronary artery disease.
5. Electrolyte Imbalance: Such as hyperkalemia, which could be visible on an abnormal heart rhythm from changes in heart rhythmpatterns.
These are general possibilities and specific diagnoses would need further clinical evaluation and possibly additional investigations such as Holter monitoring, CT angiography, a cardiac MRI, or a stress test.

**LLaVA-One-Vision-7B**

The image you've provided appears to show medical test results and an X-ray. However, since I am an AI and not a licensed medical professional, I cannot diagnose or provide a medical opinion based on these images. If you are concerned about your health or test results, you should consult with a licensed healthcare provider for a proper interpretation and advice. They would be able to evaluate the test results and the associated X-ray in the context of your symptoms and medical history, which is essential for an accurate diagnosis.

**Ground-Truth**

<think>Based on the integrated analysis of the ECG, chest X-ray (CXR), and blood test results, the findings align with the provided diagnoses of Diabetes Mellitus and Hypertension.
**Diabetes Mellitus**: **The elevated blood glucose level, while not definitively diagnostic on its own, suggests potential impaired fasting glucose**. This could indicate underlying diabetes or prediabetes, particularly in the absence of detailed glucose tolerance or HbA1c data. Additionally, the mild metabolic acidosis evidenced by the low bicarbonate and elevated anion gap may point to incomplete buffering of acids, a finding consistent with diabetic ketoacidosis in uncontrolled diabetes. Though no renal dysfunction is apparent, the possibility of early metabolic derangements related to diabetes warrants consideration given these findings.
**Hypertension:** While the ECG reveals a sinus rhythm with no overt electrical abnormalities, prolonged hypertension can result in adverse structural or functional changes not immediately evident on ECG. However, CXR findings suggest some correlation with hypertensive pathology. **The stable heart and mediastinal contours, coupled with the noted small left pleural effusion, may reflect subtle signs of longstanding hypertension potentially leading to vascular or cardiac strain over time**. Additionally, the subtle branching opacities in the right lower lobe may represent congestion secondary to hypertensive changes, though more definitive evidence of heart failure is absent. **Furthermore, the normal kidney-related parameters (creatinine, urea nitrogen) on the blood test suggest preserved renal function, which is crucial in chronic hypertension management.**
In summary, elevated glucose and metabolic abnormalities strongly support Diabetes Mellitus, while combined CXR and blood test data suggest secondary effects or subtle complications associated with Hypertension. Careful monitoring and targeted interventions for both conditions are warranted.</think>
<answer>**Diabetes Mellitus; Hypertension**</answer>

Figure 8: Performance of various MLLMs in handling medical reasoning and diagnosis. MedTVT-R1 approaches from the perspective of illness, tracing and identifying evidence of related diseases using the provided ECG, CXR, and LAB information. Qwen2.5-VL-3B-Instruct can analyze different modalities but lacks accuracy and cannot summarize diseases. InternVL3-1B combines modality information to determine disease types, though its accuracy is lacking. LLaVA-One-Vision-7B refuses to answer questions.

## G  THE USE OF LLMS

LLMs were utilized in this work for grammar checking and sentence refinement, aiming to enhance the clarity and readability of the text.

