# OpenReview forum: "MedTVT-R1: A Multimodal LLM Empowering Medical Reasoning and Diagnosis"
_ICLR.cc/2026/Conference — ICLR 2026 Conference Withdrawn Submission_

### Official Review · Reviewer_TUqC · 2025-10-26

**Soundness:** 2
**Presentation:** 3
**Contribution:** 2
**Rating:** 2
**Confidence:** 4

**Summary:**

This paper proposes a new framework for handling medical multi-modal data, which is an important research area. The work has potential for future contributions to the field. The following comments will primarily focus on the details of the experimental design used to validate the method, as outlined in the 'Weaknesses' and 'Questions' section.

**Strengths:**

The problem this paper addresses is clinically relevant and important, tackling significant research gaps. The MedTVT-QA dataset, which is claimed to be the first of its kind, is a valuable contribution that could serve as a foundation for future related work. The use of GRPO increases the novelty of this work. Overall, the writing is clear and easy to follow.

**Weaknesses:**

The experimental design has potential issues that warrant clarification, particularly concerning the choice of baselines and the unusual training/testing data split. The persuasiveness of the claimed effectiveness relies on addressing these aspects; please see details in the "Questions" section.

**Questions:**

1. Regarding Baseline Comparison Fairness: The paper states that ECG signals were converted to images and LAB data to text for baseline model evaluation. However, this conversion likely introduces significant information loss, potentially providing the baseline models with a simplified or degraded version of the input compared to the proposed MedTVT-R1, which uses specialized encoders for native data formats. Could the authors elaborate on why this comparison is considered fair despite the potential information loss? Furthermore, would it be possible to include comparisons against baseline methods adapted or designed to process these data types in their original formats (time-series for ECG, tabular for LAB) to provide a more direct benchmark?

2. Regarding Data Split and Generalization Confidence: The dataset split utilizes 8,331 samples for training and only 375 for testing (approximately 96% / 4%) out of 8,706 total combinations. Could the authors provide the rationale for this highly skewed split? While large models benefit from extensive training data, a test set of only 375 samples raises concerns about the reliability and generalizability of the reported results. Would the authors consider alternative validation strategies, such as cross-validation, or report results across multiple random splits with potentially larger test proportions, to strengthen the claims about the model's generalization capability?

---

### Official Review · Reviewer_WxUg · 2025-10-28

**Soundness:** 2
**Presentation:** 3
**Contribution:** 2
**Rating:** 4
**Confidence:** 4

**Summary:**

This work provides a new multimodal medical dataset, MedTVT-QA, which integrates ECG, CXR, and LAB data, and proposes a multimodal clinical reasoning LLM, MedTVT-R1. The model introduces a Modality Perception Layer for cross-modal interaction and integration, along with Reinforcement Fine-Tuning (RFT) to enhance reasoning capability. The proposed framework is comprehensively evaluated with both general-purpose and domain-specific foundation models, demonstrating superior performance.

**Strengths:**

1. An innovative modality interaction and integration framework for LLMs to understand multimodal EHRs.

2. A newly constructed multimodal dataset based on MIMIC for multimodal reasoning tasks.

3. A well-organized pipeline for dataset generation and model training.

**Weaknesses:**

1. In EHRs, especially those derived from the MIMIC datasets, clinical notes are a crucial modality that reflects patients’ health states. The authors should clarify why this modality was not included.

2. The paper lacks citations to prior multimodal medical reasoning studies, such as RAIM [1], ClinRaGen [2], etc.

3. The dataset only covers a small number of diseases. The authors should explain the rationale behind selecting only Coronary Artery Disease, Acute Renal Failure, Hypertension, Atrial Fibrillation, Pneumonia, Diabetes Mellitus, and Sepsis (and their subtypes).

4. Reproducibility is a concern due to insufficient data and code availability details.

5. The authors should indicate which GPT-4o API or deployment setup was used for data generation to ensure transparency and reproducibility.

6. Synthetic data reliability: LLM-generated reasoning can contain hallucinations, and further discussion or validation is needed to ensure the accuracy and credibility of the generated dataset.

References:
1. Xu, Yanbo, et al. "Raim: Recurrent attentive and intensive model of multimodal patient monitoring data." Proceedings of the 24th ACM SIGKDD international conference on Knowledge Discovery & Data Mining. 2018.

2. Shuai Niu, et al. Knowledge-Augmented Multimodal Clinical Rationale Generation for Disease Diagnosis with Small Language Models. In Proceedings of the 63rd Annual Meeting of the Association for Computational Linguistics (Volume 1: Long Papers), pages 11011–11024, Vienna, Austria. Association for Computational Linguistics. 2025

**Questions:**

Please see my concerns.

---

### Official Review · Reviewer_toHU · 2025-10-29

**Soundness:** 3
**Presentation:** 3
**Contribution:** 3
**Rating:** 4
**Confidence:** 4

**Summary:**

This paper proposes MedTVT-R1—a multimodal large language model (MLLM) for interpretable multi-disease diagnosis—by integrating three clinical modalities: ECG (time-series), CXR (visual), and LAB (tabular) data. To address the limitation of single-modal medical AI, the authors construct MedTVT-QA, which includes physiological-level QA (for modality understanding) and disease-level QA.
MedTVT-R1 features a Modality Perception Layer (MPL) (Cyclic Multi-Head Attention for cross-modal interaction, Contribution-Aware Operator for adaptive weighting) and a three-stage training pipeline: Pre-training (PT) on physiological QA, Supervised Fine-Tuning (SFT) on disease QA, and Reinforcement Fine-Tuning (RFT) via GRPO with Jaccard Reward. Experiments show it outperforms general-purpose and medical-specific MLLMs.

**Strengths:**

1.The proposed MedTVT-R1 framework effectively integrates ECG, CXR, and laboratory data, demonstrating a well-designed multimodal large language model (MLLM) architecture that addresses the inherent limitations of single-modality approaches.
2.The introduction of the MedTVT-QA dataset with a Chain of Evidence (CoE) structure represents a meaningful contribution, enabling reasoning over physiological processes and multi-disease diagnosis in a structured and interpretable manner.
3.The use of Reinforcement Fine-Tuning (RFT) with Group Relative Policy Optimization (GRPO) and a tailored Jaccard Reward demonstrates methodological innovation, effectively improving reasoning accuracy and diagnostic consistency.

**Weaknesses:**

1.The methodological innovation of this paper appears limited, as the proposed framework seems to be a combination or extension of existing approaches rather than a fundamentally novel contribution.
2. The paper does not include comparative experiments with the reward function used in DeepSeek-R1. To more convincingly support the claimed advantages, it is recommended that the authors include corresponding comparative studies.
3. The adaptive weighting fusion introduces modality bias, causing the model to over-rely on dominant modalities like ECG and overlook complementary evidence from CXR or LAB, which may lead to missed diagnoses in multi-disease scenarios. The analysis of failure cases in the paper only stays at the level of qualitative speculation and does not analyze the root causes of errors through targeted experiments.

**Questions:**

1.In Section 3.2.1, the paper mentions a learnable matrix. It is unclear whether this matrix is initialized randomly or designed based on specific prior knowledge. The authors are encouraged to clarify the initialization strategy and underlying rationale in this section.
2. What is the weight ratio of the format reward to the Jaccard reward?

---

### Official Review · Reviewer_dJyq · 2025-11-03

**Soundness:** 3
**Presentation:** 2
**Contribution:** 3
**Rating:** 6
**Confidence:** 4

**Summary:**

The paper introduces MedTVT-R1, a mutlimodal LLM model designed to integrate electrocardiogram (ECG) time series, chest X-ray (CXR) images, and laboratory test (LAB) tabular data for multi-disease diagnosis and reasoning. The authors construct MedTVT-QA, a new instruction dataset with physiological-level and disease-level question-answer pairs generated using GPT-4o with Chain of Evidence (CoE) prompting. The proposed model incorporates a Modality Perception Layer (MPL) with Cyclic Multi-Head Attention and Contribution-Aware Operator for cross-modal fusion, and employs Group Relative Policy Optimization (GRPO) with a Jaccard reward function for reinforcement fine-tuning. Experiments demonstrate superior performance compared to general-purpose and medical-domain MLLMs on both physiological understanding and disease diagnosis tasks.

**Strengths:**

- The three-stage training strategy (PT, SFT, RFT)  progressively builds model capabilities
- The Modality Perception Layer design with CMHA and CAO is technically justified for handling heterogeneous modalities
- The use of GRPO with Jaccard reward is appropriate for the multi-label disease prediction task
- The paper includes good comparisons with multiple baselines (8 general-purpose + 3 medical-specific MLLMs)

**Weaknesses:**

- The GRPO training uses only 500 iterations, which might be insufficient for convergence
- Single dataset validation (MIMIC-IV) to assess generalization
- Small model (1B) is used in the experiment

**Questions:**

- How do you handle missing values/modalities during training and inference?
- During curation, what percentage of GPT-4o generated responses were rejected or modified?
- Have you considered converting ECG signals to text/tabular data instead of  images ?

---

### Note · Authors · 2025-11-14

I have read and agree with the venue's withdrawal policy on behalf of myself and my co-authors.